# Combined Abiotic Stresses Repress Defense and Cell Wall Metabolic Genes and Render Plants More Susceptible to Pathogen Infection

**DOI:** 10.3390/plants10091946

**Published:** 2021-09-18

**Authors:** Nasser Sewelam, Mohamed El-Shetehy, Felix Mauch, Veronica G. Maurino

**Affiliations:** 1Institute of Developmental and Molecular Biology of Plants, Cluster of Excellence on Plant Sciences (CEPLAS), Heinrich Heine University Düsseldorf, Universitätsstraße 1, 40225 Düsseldorf, Germany; sewelam@science.tanta.edu.eg; 2Department of Botany and Microbiology, Faculty of Science, Tanta University, Tanta 31773, Egypt; m.shetehy@uky.edu; 3Department of Biology, University of Fribourg, 1700 Fribourg, Switzerland; felix.mauch@unifr.ch; 4Department of Molecular Plant Physiology, University of Bonn, Kirschalle 1, 53115 Bonn, Germany

**Keywords:** disease resistance, *Pseudomonas syringae*, *Botrytis cinerea*, abiotic stress, heat stress, osmotic stress, climate change

## Abstract

Plants are frequently exposed to simultaneous abiotic and biotic stresses, a condition that induces complex responses, negatively affects crop productivity and is becoming more exacerbated with current climate change. In this study, we investigated the effects of individual and combined heat and osmotic stresses on Arabidopsis susceptibility to the biotrophic pathogen *Pseudomonas syringae* pv. *tomato* (*Pst*) and the necrotrophic pathogen *B**otrytis*
*cinerea* (*Bc*). Our data showed that combined abiotic and biotic stresses caused an enhanced negative impact on plant disease resistance in comparison with individual *Pst* and *Bc* infections. Pretreating plants with individual heat or combined osmotic-heat stress strongly reduced the expression of many defense genes including pathogenesis-related proteins (*PR-1* and *PR-5*) and the *TN-13* gene encoding the TIR-NBS protein, which are involved in disease resistance towards *Pst*. We also found that combined osmotic-heat stress caused high plant susceptibility to *Bc* infection and reduced expression of a number of defense genes, including *PLANT DEFENSIN 1.3* (*PDF1.3*), *BOTRYTIS SUSCEPTIBLE 1* (*BOS1*) and *THIONIN 2.2* (*THI2.2*) genes, which are important for disease resistance towards *Bc*. The impaired disease resistance against both *Pst* and *Bc* under combined abiotic stress is associated with reduced expression of cell wall-related genes. Taken together, our data emphasize that the combination of global warming-associated abiotic stresses such as heat and osmotic stresses makes plants more susceptible to pathogen infection, thus threatening future global food security.

## 1. Introduction

The industrial revolution has increased the global atmospheric carbon dioxide concentration and the average global temperatures [1], both affecting plant physiological functions. These climatic changes also modify the interactions among plants, pathogens and their ecosystems [2]. Climate change is threatening crop yields worldwide [3,4]. For example, there is a decline in yields of important food crops at temperatures higher than 30 °C [5]. Major crops grow well at 20 °C to 25 °C. High temperatures trigger plants to develop faster, which can interfere with the proper ripening of fruits [6]. Climate change can also cause secondary adverse effects such as flooding of low-lying cropland areas because of the rising of sea level and the melting of glaciers [7]. In addition, climate change might increase the host range of pathogens by enhancing their virulence [8]. It is foreseen that the combination of abiotic and biotic stresses will occur at higher rates in the future [9].

Plants have evolved complex defense mechanisms to cope with environmental stress. However, our knowledge on how plants coordinate defense to concurrent abiotic and biotic stresses is still limited [10,11]. The simultaneous occurrence of abiotic and biotic stresses can cause a negative, neutral or positive effect on plants depending on the abiotic stress, the plant species and the pathogen [12]. The nature, strength and timing of the abiotic stresses may change the outcome of plant-pathogen interactions [13]. Exposure to combined stresses has more prominent effects than individual stresses and the stress-induced responses are not a simple sum of the effect of individual treatments [14,15,16]. Combined abiotic and biotic stresses can employ shared signals and responsive genes [17,18]. In many cases, abiotic stresses facilitate plant colonization by pathogenic microbes by significantly affecting the plant immune system through changing gene expression patterns [9,19,20]. Abiotic stresses can thus make minor pests to become potential threats in the future [21].

The environmental conditions not only affect plant responses but also pathogen behavior. Change in temperature or water availability can enhance disease development. For instance, each pathogen has an optimum temperature for growth and virulence [22]. Small temperature fluctuations can increase the susceptibility of potato against *Phytophthora infestans* [23]. High temperature was reported to increase the severity of soft rot disease caused by *Pectobacterium atrosepticum*, through increasing the production of plant cell wall degrading enzymes [24]. Similarly, high temperature suppressed the resistance of tobacco (*Nicotiana tabacum*) against Tobacco Mosaic Virus and pepper (*Capsicum annuum*) against Tomato Spotted Wilt Virus [25,26]. Furthermore, high temperature abolished both basal and resistance (R)-gene-mediated defense responses against *Pseudomonas syringae* (*Pst*) in Arabidopsis and tobacco [27]. Although elevated temperature adversely affects type III secretion of *Pst* in vitro [28], it increases type III translocation of effectors into Arabidopsis during infection [29]. On the other hand, limited availability of water such as water deficit or osmotic stress, or excess water due to flooding can negatively impact both plants and microbes simultaneously [20]. To overcome drought, plants upregulate the level of the phytohormone abscisic acid (ABA), which facilitates the closure of stomata to reduce transpiration, resulting in reduction of bacterial entry via stomata [30]. Conversely, the accumulation of ABA antagonizes salicylic acid (SA) defense signaling pathways leading to abolishing SA-mediated resistance in leaves [31,32].

Extensive efforts have been made to understand plant-microbe interactions at all levels. In contrast, the impact of unfavorable environmental conditions on host-pathogen interactions is not fully decoded. Understanding these interactions is important for predicting disease outbreaks especially with current and future climate change threats [20]. Here, we analyzed how occurrence of individual and simultaneous global warming-associated abiotic challenges such as heat and osmotic stresses alter plant-pathogen interactions. For this purpose, Arabidopsis plants subjected to individual and combined osmotic and heat stresses were infected either with the biotrophic bacteria *Pseudomonas syringae* pv. *Tomato* (*Pst*) or the necrotrophic fungus *B**otrytis*
*cinerea* (*Bc*). Both pathogens were selected as they have different life styles; biotrophic pathogens feed on the nutrients of the living host tissues, while necrotrophic pathogens kill the host tissue and survive on its contents [33]. Our results indicated that combined global warming-associated abiotic stresses have synergistic negative effects on disease resistance compared with the individual stresses.

## 2. Results

### 2.1. Heat and Combined Osmotic-Heat Stresses Weaken Arabidopsis Resistance to Pseudomonas syringae (Pst)

We investigated the effects of individual and combined osmotic and heat stresses on Arabidopsis susceptibility to the biotrophic pathogen *Pst* after three days of inoculation. Under heat or combined osmotic-heat stress, the *Pst*-inoculated leaves developed severe disease symptoms in the form of chlorosis (Figure 1A). In contrast, osmotic-treated plants developed mild disease symptoms (Figure 1A). The quantification of pathogen infection after three days of inoculation indicated that the osmotic stressed plants showed no significant increase in the number of bacteria compared to the *Pst*-treated plants. Heat stress resulted in 8.2-log2 fold increase in bacterial growth in comparison to the *Pst*-treated plants (Figure 1B), while plants subjected to combined osmotic-heat stress showed higher susceptibility compared to all other treatments (Figure 1B).

### 2.2. Combined Osmotic-Heat Stress Highly Enhances Disease Susceptibility of Arabidopsis against Botrytis Cinerea (Bc)

To analyze whether the response of abiotic-stressed plants to a biotrophic pathogen would be different from their response to a necrotrophic pathogen, we further investigated the effects of individual and combined osmotic and heat stresses on Arabidopsis susceptibility to the necrotrophic pathogen *Bc*. After three days of inoculation, plants that were pretreated with individual osmotic or heat stresses showed similar enhanced susceptibility levels to infection compared to the control (Figure 2A). Combined osmotic-heat stress caused much higher susceptibility, represented by larger lesions than those observed in plants under the individual stresses (Figure 2A). Quantitative analysis of the infection levels indicated that the lesion sizes in plants treated with combined *Bc* and osmotic or *Bc-* and heat stresses were significantly larger than those in plants only infected with *Bc*; with 11.0- and 11.2-fold increase in lesion size compared to control, respectively (Figure 2B). Plants pretreated with combined osmotic-heat stress showed a much higher increase in the lesion size (25.9-fold) when inoculated with *BC* compared to the control plants (Figure 2B).

### 2.3. Osmotic and Heat Stresses Decrease the Expression of Many Defense and Cell Wall Related Genes

The enhanced susceptibility of Arabidopsis plants against *Pst* and *Bc* infections observed after individual and combined osmotic and heat stresses prompted us to investigate the effects of these abiotic stresses on the expression levels of Arabidopsis defense related genes using our recently performed genome-wide RNA-seq analysis [16]. Among 730 genes annotated as involved in biotic stress responses in the Arabidopsis genome, we found that 77 genes showed at least 1.5-fold repression and 37 genes showed at least 1.5-fold induction under the combined osmotic-heat treatment compared to the control (Appendix A). Table 1 shows 20 highly repressed defense-related genes taking into account the individual or the combined osmotic and heat treatments. This list includes key genes involved in defense against biotrophic pathogens such as *pathogenesis-related proteins-1* and -5 (*PR-1* and *PR-5*) and *TIR-NBS13* (*TN13*) and against necrotrophic pathogens such as plant defensin 1.2 and 1.3 (*PDF1.2* and *PDF1.3*), botrytis susceptible 1 (*BOS1*) and thionin2.2 (*THI2.2*) (Table 1).

As the plant cell walls serve as the first line of defense against pathogens [34] and to find a link between the observed plant susceptibility and cell wall metabolism under the applied abiotic stresses, we analyzed the effects of individual and combined osmotic and heat stresses on the expression levels of genes annotated as cell wall-related [16]. Out of 532 cell wall-related genes, 90 genes showed at least 1.5-fold repression and 25 genes showed at least 1.5-fold induction under the combined osmotic-heat treatment compared to control (Appendix A). These genes encode mostly for enzymes associated with cell wall metabolism. Table 2 shows 20 highly repressed cell wall-related genes taking into account the individual or the combined osmotic and heat treatments. Among the cell wall repressed genes, the *xyloglucan endotransglucosylase/hydrolases 20* and *25* (*XTH20* and *XTH25*) and *fasciclin-like arabinogalactan 2* (*FLA2*), showed pronounced levels of repression under the applied abiotic stresses. *XTH20* and *XTH25* belong to the XTH gene family and *FLA2* belongs to the FLA gene family, both including many members playing crucial roles in the synthesis and integrity of plant cell walls.

### 2.4. Heat Stress Dominates the Inhibition of Transcriptional Response of Defense Genes against Pst Infection

To further analyze the link between the impairment of disease resistance and genes involved in the defense response to the biotrophic bacteria *Pst*, we quantified the changes in expression levels of selected genes known to be involved in plant responses to biotrophic pathogens. Based on the data presented in Table 1, we selected the salicylic acid-responsive marker genes, *PR-1* and *PR-5* and *TN13* encoding TIR-NBS protein for further analysis. Our results showed that the expression levels of *PR-1*, *PR-5* and *TN13* were highly induced after *Pst* infection with 11.5-, 1.6- and 2.5- log2 fold enhancement, respectively, compared to mock-treated plants (Figure 3A–C). Application of osmotic stress before infection with *Pst* had no considerable effect on the induction of *PR-1* and *PR-5* (Figure 3A,B). In contrast, osmotic stress caused 1.0-log2 fold reduction of *TN-13* expression in comparison with mock-treated plants and an induction of 0.75-log2 fold after *Pst* inoculation, which represents 1.8-log2 fold less induction of expression than in plants treated with *Pst* alone (Figure 3C).

On the other hand, the expression levels of *PR-1* and *PR-5* were reduced under all conditions that include a pre-treatment with heat stress compared to the individual *Pst* treatment (Figure 3A–C). Although treatment with *Pst* resulted in 5.6-log2 fold increase in transcript levels of *PR-1* in heat stressed plants, the induction was 5.7-log2 fold less than *Pst*-stressed plants (Figure 3A). Treatment with *Pst* could not recover the repressed gene expressions in heat-stressed plants where plants showed 0.75- and 2.3-log2 fold decrease in transcript abundance of *PR-5* and *TN-13*, respectively, in comparison with mock-treated plants (Figure 3B,C). In addition, combined osmotic-heat stress led to 2.5, 0.72- and 2.1-log2 fold reduction in transcript abundance of *PR-1*, *PR-5* and *TN-13*, respectively, relative to mock-treated plants (Figure 2A–C). Importantly, pretreating plants with combined osmotic and heat stresses led to a higher repression of the three investigated genes after infection with *Pst* (Figure 3A–C).

### 2.5. Combined Osmotic-Heat Stress Downregulate the Expression of Defense Related Genes against Bc Infection

We analyzed the effect of combined abiotic and biotic stresses on the transcript levels of selected defense genes involved in resistance against the necrotrophic fungus *Bc*. Based on the data presented in Table 1, we selected *PDF1.3*, *BOS1* and *THI2.2* for further analysis. Our data show that osmotic stress caused a significant decrease in the expression levels of *PDF-1.3*, *BOS1* and *THI2.2* compared to control plants (Figure 4A–C). The effect of osmotic stress on *PDF-1.3* and *BOS1* expression was partially compensated after *Bc* infection (Figure 4A,B). Combined osmotic-*Bc* stress caused 2.5- and 0.3-log2 fold induction of *PDF-1.3* and *BOS1* expression in comparison with mock-treated plants, which represent 2.0- and 0.68-log2 fold less than plants infected only with *Bc* (Figure 4A,B). Interestingly, combined osmotic-*Bc* treatment was not able to rescue the negative effect of osmotic stress alone on the expression levels of *THI2.2* where it showed 4.0-log2 fold reduction of expression compared to mock-treated plants (Figure 4C).

Heat treatment caused 3.4-, 0.49- and 3.2-log2 fold reduction of expression of *PDF-1.3*, *BOS1* and *THI2.2*, respectively, compared to mock-treated plants (Figure 4A–C). The reduction of expression of *PDF-1.3* and *BOS1* was compensated when heat-stressed plants were challenged with *Bc*; these plants showed 3.5- and 1.4-log2 fold increase in expression of *PDF-1.3* and *BOS1,* respectively, in comparison with mock-treated plants (Figure 4A–C). Conversely, the expression of *THI2.2* was significantly reduced in combined heat-*Bc*-stressed plants, which showed 5.8-log2 fold reduction of expression compared to mock-treated plants (Figure 4C). Combined osmotic-heat stress treatment led to a remarkable decrease of expression of all tested genes compared to control plants or individual *Bc*-stressed plants (Figure 4A–C). Infecting the osmotic-heat-stressed plants with *Bc* did not reverse the reduction of expression of any of the analyzed genes (Figure 4A–C).

### 2.6. Impaired Disease Resistance against Both PST and BC under Combined Abiotic Stress Is Associated with Reduced Expression of Cell Wall-Related Genes

Among cell wall-related genes in Arabidopsis, the XTH gene family (includes 33 members) and the FLA gene family (includes 20 members) are involved in the synthesis and integrity of plant cell walls [16]. Based on data presented in Table 2, we selected two members out of these big gene families: *XTH20* and *FLA2* for further analysis of their expression under individual and combined abiotic and biotic stress treatments. Our results showed that all individual abiotic and combined abiotic-biotic treatments led to a significant negative effect on the expression of *XTH20* in comparison with the mock-treated plants and plants infected with *Pst* alone. The highest downregulation of *XTH20* expression was observed in plants under combined osmotic-heat-*Pst* stress, with a strong reduction of 12.8- and 11.9-log2 fold compared to the control and individual *Pst*-inoculated plants, respectively (Figure 5A). Treating Arabidopsis plants with osmotic, heat and combined osmotic-heat stresses caused a significant reduction in expression levels of *FLA2* after infection with *Pst* and reached a reduction of 9.5-, 8.1- and 9.6-log2 fold compared to the mock-treated plants, respectively (Figure 5B).

On the other hand, individual abiotic stresses and their combination strongly repressed the transcript levels of *XTH20* when plants were inoculated with *Bc* compared to the mock-treated control and *Bc*-inoculated plants (Figure 5C). The expression levels of *FLA2* after osmotic, heat and osmotic-heat treatments showed 5.4-, 1.8- and 2.8-log2 fold reduction in comparison to the mock-treated plants, respectively (Figure 5D). In addition, osmotic-*Bc*, heat-*Bc* and osmotic-heat-*Bc* stress combinations led to a decrease in the transcript levels of *FLA2* compared to the mock-treated and the *Bc*-inoculated plants (Figure 5D).

## 3. Discussion

In the current study, we investigated the responses of Arabidopsis plants treated with individual and combined abiotic stresses to infection with the biotrophic bacterial pathogen *Pst* and the necrotrophic fungal pathogen *Bc*. The data presented here showed that individual and combined osmotic and heat stresses caused a higher susceptibility to infection by both types of pathogens, which parallels a general downregulation of the expression of many defense and cell wall-related genes.

Our results indicated that heat and more importantly the combination of heat with osmotic stresses enhanced the susceptibility of Arabidopsis plants to *Pst* infection, which was accompanied by the downregulation of expression of many defense-related genes (Figure 1 and Figure 3 and Appendix A and Table 1). Individual heat stress was reported to abolish both basal and resistance (R)-gene-mediated defense responses and pathogen-triggered immunity signaling and resistance to *Pst* DC3000 in Arabidopsis plants [27,35]. Heat stress can cause enhanced basal disease susceptibility of Arabidopsis to *Pst* by increasing type III translocation of effectors into host plants [29,36].

We further investigated the effect of the combined global warming-associated abiotic stresses on the plant transcriptional response of selected defense-related genes to *Pst*. We found that the transcript levels of two salicylic acid-responsive marker genes, *PR-1* and *PR-5*, encoding the pathogenesis-related protein 1 (PR-1) and pathogenesis-related protein 5 (PR-5), were substantially repressed in response to heat stress or the combined osmotic-heat stress (Figure 3A,B). Reduction of the activity of PR-1 under high temperature was attributed to a loss of isochorismate synthase 1 (ICS1)-mediated SA biosynthesis [29]. We also observed that the enhanced expression level of *TN-13* by *Pst* infection was greatly reversed when the treatment with the pathogen was preceded by any of the abiotic stress treatments individually or in combination (Figure 3C). TN-13 is involved in resistance against *Pst* where it acts as a component of plant innate immunity that binds to MOS6/IMPORTIN in response to pathogen stimulus [37]. Thus, our results suggest an impaired defense response against *Pst* due to a general decrease in expression of defense-related genes after the abiotic treatments.

We found that plants treated with individual osmotic or heat stresses showed similar enhanced disease susceptibility to the necrotrophic fungus *Bc* compared to non-stressed plants. Notably, combined osmotic-heat stress significantly enhanced disease susceptibility to *Bc* compared to the individual stresses (Figure 2). This enhanced susceptibility of Arabidopsis to *Bc* infection after osmotic treatment, which can be used as a proxi of drought, contrasts with the proposed role of drought in enhancing resistance of tomato against *Bc* by increasing ABA levels [38]. The observed opposing effects on disease resistance could be species related or due to differences in stress application (osmotic vs watering withholding) and infection assays. Our data showed that the expression levels of *PDF1.3* and *BOS1* were reduced in comparison with mock-treated plants under the individual and combined abiotic stresses.

Interestingly, combined osmotic-heat stress followed by an inoculation of *Bc* did not recover the induction of the transcript levels of both genes compared to *Bc* treated plants (Figure 4A,B). The accumulation of plant defensins plays key roles in mediating jasmonic acid/ethylene defense response against necrotrophic pathogens such as *Bc* [39]. *BOS1* encodes a R2R3MYB transcription factor, which is known to mediate plant responses to both biotic and abiotic stresses. In this regard, *Bos1* mutant plants showed increased susceptibility to *Bc* [40]. Additionally, many reports demonstrated that biotic- and abiotic stress-inducible antimicrobial thionins are involved in plant defense [41,42]. This is compatible with our results showing that the individual and combined abiotic stresses caused a high significant reduction in the expression level of the *THI2.2* gene compared to the controls (Figure 4C). Together, these results suggest that the enhanced disease susceptibility in Arabidopsis plants against *Bc* under combined abiotic stresses (Figure 2) is associated with their inability to induce defense-related genes against necrotrophic pathogen attack, the negative effect of the double abiotic stress overriding the defensive response to biotic stress.

As plant cell walls represent the first defense line against pathogen infections, we studied the effects of individual and combined stress treatments on the expression of genes annotated as cell wall-related in Arabidopsis. Individual and combined abiotic stresses highly decreased the expression of cell wall-related genes, among them *XTH20* and *FLA2* (Appendix A, Table 2 and Figure 5A–D). Xyloglucan endotransglucosylase/hydrolase (*XTH*), including XTH20, have diverse functions in cell enlargement and modification and reconstruction of the cell wall network [43]. On the other hand, Fasciclin-like arabinogalactan proteins (FLAs) are likely involved in cell wall integrity [44] and the regulation of cell wall biosynthesis [45]. Our data showed that the expression levels of *XTH20* and *FLA2* were highly reduced under all individual and combined abiotic stresses after pathogen infection, while we observed induction of the expression of *XTH20* in plants that were only infected with *Bc* (Figure 5B,D). *XTH20* expression is regulated by different NAC transcription factors (*ANAC07*, *ANAC019* and *ANAC055*) [43,46]. Downstream target genes of ANAC07 have been reported to play a role in abiotic dehydration stress responses, secondary wall biosynthesis and defense responses [43,46]. Thus, the transcriptional changes we observed probably result from the described JA-induced expression of defense genes against *Bc* infection through *ANAC019* and *ANAC055* [43,46]. Together, these data suggest that the repressed expression of cell wall-related genes under individual and combined abiotic stresses most probably play an additive role in impairing disease resistance against both *Pst* and *Bc*.

In conclusion, our work indicates that the combination of global warming-associated abiotic stresses such as heat and drought strongly impair disease resistance against *Pst* and *Bc* shedding further light on the negative impact climate change have on plant disease resistance. Future work should focus on the analysis of mutants and overexpressing lines from the selected genes, such as those analyzed in this work, to shed light on the role of these genes under abiotic stress and to further deepen our understanding on the nature of the interactions between combined abiotic and biotic stresses. In addition, this study highlights the need for future studies to predict the severity of different climate change scenarios on plant responses to combined abiotic and biotic stresses and on crop plant productivity.

## 4. Materials and Methods

### 4.1. Growth Conditions and Abiotic Stress Treatments

Seeds of *A. thaliana* wild type Col-0 were sown on soil and stratified at 4 °C for 2 days. Seedlings at the four-leaf-stage were transplanted to new soil. The plants were kept in the normal growth chamber with a 16 h light/8 h dark photoperiod and 22 °C/18 °C day/night temperature. Four-week-old plants were separated into four groups and treated as follows. As a proxy for drought stress, osmotic stress treatment was applied to a group of plants by watering with a mannitol solution (200 mM) and left for 16 h according to the protocol of Sewelam et al. [14,16] before inoculation of half of these plants with *Pseudomonas syringae* pv. *tomato* (*Pst*) DC3000 or *Botrytis*
*cinerea* (*Bc*). For heat stress treatment, a second group of plants was transferred to a growth chamber with a 16 h light/8 h dark photoperiod and 31 °C/27 °C day/night temperature and left for 4 h before pathogen inoculation of half of these plants. For combined osmotic-heat stress, a third group of plants were watered with mannitol solution (200 mM) and after 12 h were transferred to a growth chamber of 31 °C/27 °C day/night temperature and left for further 4 h before pathogen inoculation of half of these plants [14,16]. A fourth group of plants was kept in the normal growth chamber and used as control of abiotic stress. The experiments of infection with *Pst* and *Bc* were conducted separately and in three biological replicates.

### 4.2. Inoculum Preparation and Biotic Stress Treatments

*Pst* DC 3000 (provided by Prof. Markus Geisler, University of Fribourg, Switzerland) was prepared by inoculating a single bacterial colony in 10 mL of King’s B medium (1.5 g K_2_HPO_4_, 1.5 g MgSO_4_.7H_2_O, 20 g tryptone, 10 mL glycerol per 1 L of water) containing rifampicin (25 μg/mL). After overnight incubation on a shaker at 28 °C in the dark, the cells were centrifuged at 3000 rpm for 10 min and the pellet was suspended in 10 mM MgCl_2_. The optical density was measured at A600. Four leaves were infiltrated with MgCl_2_ or *Pst*. The inoculated plants were transferred to the growth chambers.

*Bc* strain BMM (provided by Prof. Brigitte Mauch-Mani, University of Neuchâtel, Switzerland) was grown on potato dextrose agar (PDA) plates. Spores were harvested in water and filtered via glass wool to get rid of hyphae. Spores were diluted in 1:4 strength potato dextrose broth (PDB) for inoculation. The plants were inoculated by the drop method (10^6^ spores per mL, 20 μL droplet/leaf). To assure high humidity conditions, the inoculated plants were kept in covered trays. Control plants were mock inoculated with 1:4 strength PDB solution.

### 4.3. Quantification of Pathogen Infection

For quantification of *Pst*, leaf discs (4 mm) were harvested from the inoculated leaves at 0- and 3-day post inoculation (dpi). The leaf discs were homogenized in 10 mM MgCl_2_ and the undiluted (0 dpi) or the 10^3^-fold diluted (3 dpi) homogenates were plated on King’s B agar plates. The plates were incubated at 28 °C in the dark for 48 h. Then, the bacterial colonies were counted and quantified as colony forming unit (CFU) per leaf disc [47]. For quantifying *Bc*, disease symptoms of inoculated plants were quantified by measuring the lesion size at 3 dpi. The lesion size on the drop-inoculated leaves was measured using a digital Mahr caliper [48].

### 4.4. Real Time RT-PCR Analysis

Arabidopsis leaf samples were collected 48 h after mock treatments or pathogen infections, frozen in liquid nitrogen and stored at −80° C. After grinding samples to a fine powder, total RNA was extracted from 100 mg of powder with the Spectrum Plant Total RNA Kit (Sigma Life Science, St. Louis, MI, USA). One microgram of total RNA was used for cDNA synthesis using the Omniscript Reverse Transcription Kit (Qiagen, Germany; Catalog No.205113). The reaction mixture for RT-qPCR contained 7.5 μL of 2x SensiMix^TM^ SYBR Hi-ROX Mastermix (Bioline, Meridian Bioscience, UK; Catalog No. QT605-05), 5 μL of cDNA (corresponding to 25 ng RNA) and primers at a concentration of 10 μM in a final volume of 15 μL. Runs were performed on a MIC qPCR machine using micPCR v2.8.13 analysis program. The final qRT-PCR products were analyzed by melting point analysis. Transcript levels of *PR-1*, *PR-5*, *TN-13*, *PDF1.3*, *BOS1*, *THI2.2*, *FLA2* and *XTH20* in Arabidopsis plants were calculated with the plant biomass reference gene *expG* [49] and the comparative cycle threshold method (DDCt). The used gene-specific primers are listed in Appendix A.

### 4.5. Statistical Analysis

Statistical analysis was performed for a minimum of three biological replicates. Data are represented as mean ± SD. Statistical analysis was performed using IBM SPSS Statistics (version 22) using one-way ANOVA with post hoc tests.

## Figures and Tables

**Figure 1 plants-10-01946-f001:**
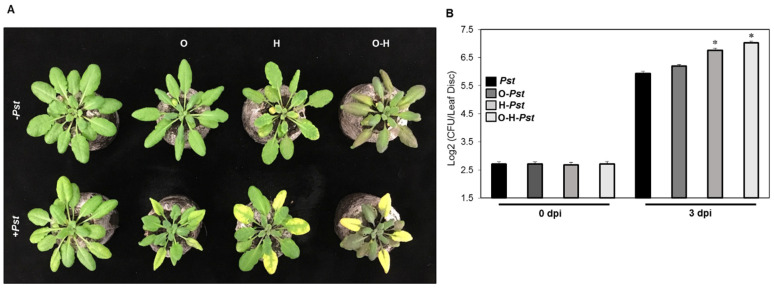
Effect of individual and combined osmotic and heat stresses on plant susceptibility to *Pseudomonas syringae* pv. tomato DC3000 (*Pst*). (**A**) Phenotype of stressed Arabidopsis (Col-0) plants. Plants were infected with *Pst* 16 h after the abiotic stress treatments. Photos were taken 3 d after *Pst*-inoculation. (**B**) Bacterial growth on leaves, quantified at 0 and 3 dpi (days post-infection) and expressed as Log2 values of colony forming units (CFU). O: osmotic stress, H: heat. Bars represent the mean and standard deviation of three biological replicates. Asterisks indicate significant differences, 1-way ANOVA; post-hoc LSD, * *p* < 0.05.

**Figure 2 plants-10-01946-f002:**
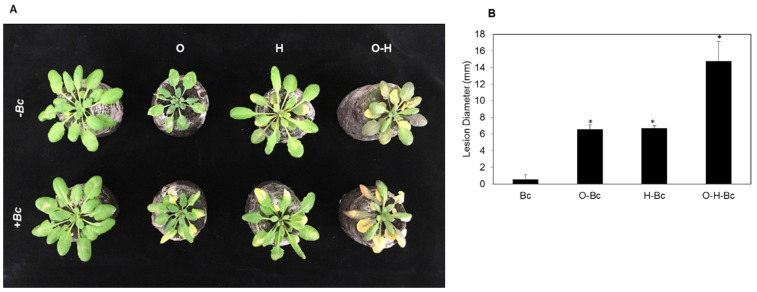
Effect of individual and combined osmotic and heat stresses on plant susceptibility to *Botrytis cinerea* (*Bc*). (**A**) Phenotype of stressed Arabidopsis (Col-0) plants. Plants were infected with *Bc* 16 h after the abiotic stress treatments. Photos were taken 3 d after *Bc*-inoculation. (**B**) Infection degree as lesion size on leaves, measured 3 dpi (days post-infection). O: osmotic stress, H: heat. Bars represent the mean and standard deviation of three biological replicates. Asterisks indicate significant differences, 1-way ANOVA; post-hoc LSD, * *p* < 0.05.

**Figure 3 plants-10-01946-f003:**
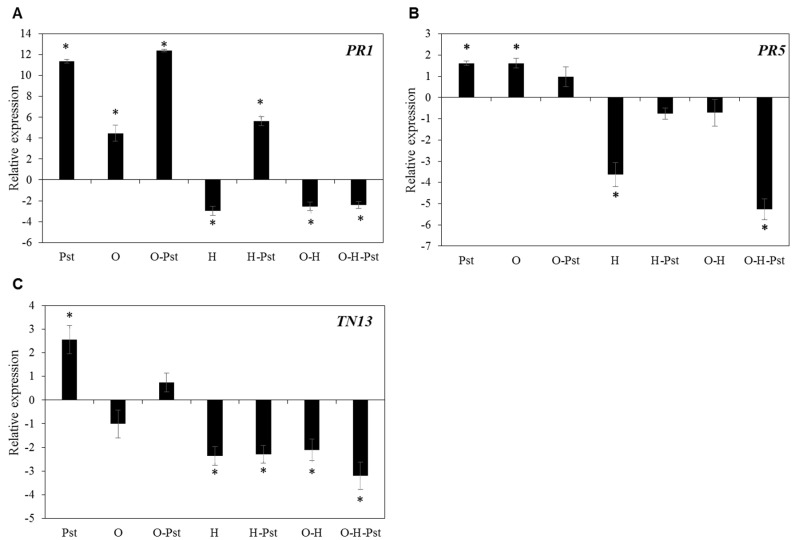
Effect of individual and combined osmotic, heat and *Pseudomonas syringae* pv. tomato DC3000 (*Pst*) treatments on the expression of *AtPR-1* (**A**), *AtPR-5* (**B**) and *AtTN-13* (**C**) genes in Arabidopsis (Col-0) plants. *Pst* was applied 16 h after the abiotic stress treatments. Leaves were sampled 48 h after MgCl_2_ (mock) or *Pst* treatments. O: osmotic stress, H: heat. Data are represented as log2 fold change, normalized with reference gene (plant biomass expressed protein, expG) and relative to the mock-treated control. The zero-line represents MgCl_2_-infiltrated (mock) plants. Bars represent the mean and standard deviation of three biological replicates. Asterisks indicate significant differences, 1-way ANOVA; post-hoc LSD, * *p* < 0.05.

**Figure 4 plants-10-01946-f004:**
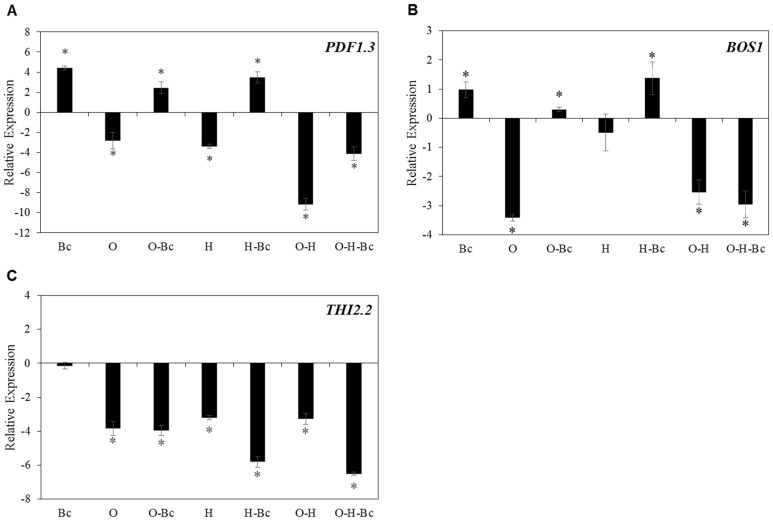
Effect of individual and combined osmotic, heat and *Botrytis cinerea* (*Bc*) treatments on the expression of *AtPDF1-3* (**A**), *AtBOS1* (**B**) and *AtTHI2.2* (**C**) genes in Arabidopsis plants (Col-0). *Bc* was applied 16 h after the abiotic stress treatments. Leaves were sampled 48 h after ¼ strength PDB solution (mock) or *Bc* treatments. O: osmotic stress, H: heat. Data are represented as log2 fold change, normalized with reference gene (plant biomass expressed protein, expG) and relative to the control. The zero-line represents plants mock-treated with ¼ strength PDB solution. Bars represent the mean and standard deviation of three biological replicates. Asterisks indicate significant differences, 1-way ANOVA; post-hoc LSD, * *p* < 0.05.

**Figure 5 plants-10-01946-f005:**
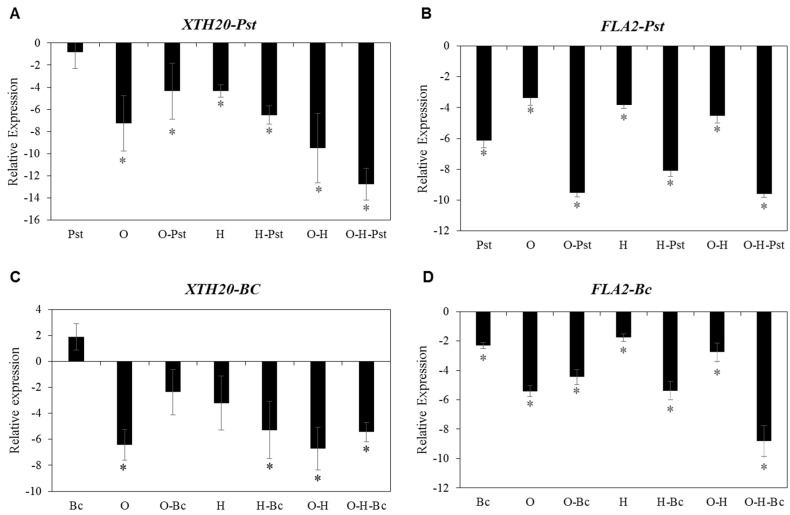
Effect of individual and combined osmotic, heat, *Pseudomonas syringae* pv. tomato DC3000 (*Pst*) (**A**,**B**) or *Botrytis cinerea* (*Bc*) (**A**,**D**) treatments on the expression of *AtXTH20* (**A**,**C**) and *AtFLA2* (**B**,**D**) genes in Arabidopsis plants (Col-0). *Pst* and *Bc* were applied 16 h after the abiotic stress treatments. Leaves were sampled 48 h after mock or pathogen treatments. O: osmotic stress, H: heat. Data are represented as log2 fold change, normalized with reference gene (plant biomass expressed protein, expG) and relative to the control. The zero-line represents mock treated plants. Bars represent the mean and standard deviation of three biological replicates. Asterisks indicate significant differences, 1-way ANOVA; post-hoc LSD, * *p* < 0.05.

**Table 1 plants-10-01946-t001:** List of 20 highly repressed defense-related genes taking into account the individual or the combined osmotic and heat treatment.

Gene Locus	Gene Name	Abiotic Stress Treatment		
		O	H	O-H
AT3G22231	PCC1 (pathogen and circadian controlled 1)	−5.16	−14.93	−27.67
AT1G33960	AIG1 (AVRRPT2-INDUCED GENE 1)	−3.99	−5.91	−8.38
AT1G65390	PP2-A5 (phloem protein 2 A5)	−2.04	−5.24	−8.12
AT1G33950	AIG1 (avirulence induced gene family protein)	−6.02	−5.06	−6.19
AT1G13609	Defensin-like (DEFL) family protein	−11.13	−2.69	−6.07
AT1G59780	NB-ARC domain-containing disease resistance protein	−2.12	−5.25	−5.42
AT3G48080	Alpha/beta-Hydrolases superfamily protein	−1.67	−2.87	−4.29
AT2G14580	PRB1 (basic pathogenesis-related protein 1)	−1.79	−1.93	−4.29
AT2G14610	PR1 (pathogenesis-related gene 1)	−3.32	−3.08	−4.10
AT5G25910	RLP52 (receptor like protein 52)	1.98	−3.12	−3.83
AT5G44430	PDF1.2c (plant defensin 1.2C)	−5.81	−4.11	−3.81
AT5G44420	PDF1.2 (plant defensin 1.2)	−6.00	−4.52	−3.69
AT3G04210	TN13 (Disease resistance protein, TIR-NBS class)	−1.43	−2.83	−3.42
AT5G24200	PR-protein, Alpha/beta-Hydrolases superfamily protein	−2.66	−5.08	−3.36
AT2G26010	PDF1.3 (plant defensin 1.3)	−8.33	−3.34	−3.24
AT3G48720	DCF (HXXXD-type acyl-transferase family protein)	−1.42	−2.94	−3.19
AT2G26020	PDF1.2b (plant defensin 1.2b)	−2.24	−3.77	−2.27
AT5G36910	THI2.2 (thionin 2.2)	−2.05	1.06	−1.96
AT1G75040	PR5 (pathogenesis-related gene 5)	3.11	−2.62	1.94
AT3G06490	MYB108 or BOS1 (Botrytis-susceptible1)	−1.82	−1.20	−1.53

O: osmotic stress; H: heat; O-H: osmotic and heat stress. Numbers indicate fold changes. The negative signs indicate repression, i.e., −27.67 means that gene expression is 27.67-fold lower under treatment. Genes are ordered from the most to the less repressed expression after combined osmotic-heat treatment.

**Table 2 plants-10-01946-t002:** List of 20 highly repressed cell wall-related genes by the individual or combined osmotic and heat treatments.

Gene Locus	Gene Name/Function	Abiotic Stress Treatment		
		O	H	O-H
AT5G57550	XTH25 (xyloglucan endotransglucosylase/hydrolase 25)	−4.94	−1.94	−10.34
AT4G13410	ATCSLA15 (encodes a gene similar to cellulose synthase)	−2.03	−6.30	−8.47
AT5G48070	XTH20 (xyloglucan endotransglucosylase/hydrolase 20)	−1.44	−8.72	−7.18
AT4G15320	CSLB06 (cellulose synthase-like B6)	−1.69	−6.94	−5.40
AT1G19940	GH9B5 (glycosyl hydrolase 9B5)	−1.16	−4.64	−4.73
AT2G20870	Cell wall protein precursor	−1.31	−2.51	−4.40
AT2G45220	Plant invertase/pectin methylesterase inhibitor	−2.43	−2.19	−3.66
AT4G12730	FLA2 (FASCICLIN-like arabinogalactan 2)	−1.51	−2.28	−3.00
AT1G14080	FUT6 (fucosyltransferase 6)	−1.43	−3.88	−2.97
AT5G06870	PGIP2 (polygalacturonase inhibiting protein 2)	−2.03	−1.98	−2.85
AT4G13210	Pectin lyase-like superfamily protein	−1.07	−2.62	−2.82
AT3G27400	Pectin lyase-like superfamily protein	−1.06	−3.16	−2.80
AT5G26670	Pectinacetylesterase family protein	−1.20	−2.41	−2.73
AT2G26440	Plant invertase/pectin methylesterase inhibitor	−0.81	−2.46	−2.72
AT4G01630	EXPA17 (expansin A17)	−0.98	−4.18	−2.71
AT5G45280	Pectinacetylesterase family protein	−1.14	−2.00	−2.69
AT2G04780	FLA7 (FASCICLIN-like arabinoogalactan 7)	−1.11	−2.27	−2.35
AT4G24780	Pectin lyase-like superfamily protein	−1.06	−2.20	−1.96
AT1G24070	CSLA10 (cellulose synthase-like A10)	−1.37	−2.18	−1.68
AT1G35230	AGP5 (arabinogalactan protein 5)	2.88	−2.78	−1.31

O: osmotic stress; H: heat; O-H: osmotic and heat stress. Numbers indicate fold changes. The negative signs indicate repression, i.e., −10.34 means that gene expression is 10.34-fold lower under treatment. Genes are ordered from the most to the less repressed expression after combined osmotic-heat treatment.

## Data Availability

The data presented in this study are available in [16] and Appendix A here.

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
