# Peer review of "Combined Abiotic Stresses Repress Defense and Cell Wall Metabolic Genes and Render Plants More Susceptible to Pathogen Infection"

_plants, 2021, doi:10.3390/plants10091946_

Round 1

Reviewer 1 Report

The study presents results from bioassays using biotrophic and necrotrophic pathogens on Arabidopsis plants under individual or combined heat or osmotic (drought) stress. The findings show that heat stress, and to a lesser degree, osmotic stress impair the ability of Arabidopsis to defend against both classes of pathogens. Combined heat and osmotic stress exacerbated the susceptibility. The paper also conducts qPCR for a small number of genes with roles in defense responses and cell wall formation and makes the case that transcriptional responses to abiotic stress leave plants more vulnerable to pathogen infection. These findings add incrementally to our general understanding of the interactions between biotic and abiotic stresses in Arabidopsis and are not surprising.

The paper is well-written and the experiments are designed properly, conducted appropriately, and presented carefully. The statistics are done correctly, though I recommend a more thorough analysis that could provide more strength to interpretations of the qPCR data. I have no major concerns with the paper, though I do have a number of minor points that should be addressed, which follow:

  1. Adding a table summarizing a global analysis of the two categories of genes you target (Involved in Biotic Stress Responses) and (Cell Wall Related) would be informative. For example, how many of the 730 and 532 genes annotated as each of the two categories were decreased by abiotic stresses? How many increased?  To me, this might strengthen claims made in the paper that may not be supported with the limited presentation currently in the paper.
  2. Selecting a few genes to tests with qPCR out of a list of genes from an RNAseq paper that are known to be down-regulated seems like cherry-picking. Making broad claims like “parallels a general downregulation of the expression of many defense and cell-wall related genes.” (Line 292), may not be warranted just based on the genes listed in Tables 1 and 2 plus the qPCR data. My Suggestion #1 may help address this.
  3. I strongly suggest conducting statistical analysis between each treatment/partition on the qPCR data, not only comparting each treatment against the NT control, but also between each other. For example, is there a significant difference between Pst-treated and osmotically-stressed Pst-treated?

I think these suggestions would strengthen the paper’s conclusions. However, if the authors have good reasons for not doing these, I think the paper is fit for publication without them.

Author Response

Point 1: Adding a table summarizing a global analysis of the two categories of genes you target (Involved in Biotic Stress Responses) and (Cell Wall Related) would be informative. For example, how many of the 730 and 532 genes annotated as each of the two categories were decreased by abiotic stresses? How many increased?  To me, this might strengthen claims made in the paper that may not be supported with the limited presentation currently in the paper.

Response: In the revised manuscript we now indicate the number of genes showing at least 1.5-fold repression or induction under the combined osmotic-heat treatment compared to control (lines 143-145 and 159-162). These genes are highlighted in blue (repression) and red (induced) in the most left column in the Supplementary Tables S1 and S2.

Point 2: Selecting a few genes to tests with qPCR out of a list of genes from an RNAseq paper that are known to be down-regulated seems like cherry-picking. Making broad claims like “parallels a general downregulation of the expression of many defense and cell-wall related genes.” (Line 292), may not be warranted just based on the genes listed in Tables 1 and 2 plus the qPCR data. My Suggestion #1 may help address this.

Response: Please refer to response of point #1.

Point 3: I strongly suggest conducting statistical analysis between each treatment/partition on the qPCR data, not only comparting each treatment against the NT control, but also between each other. For example, is there a significant difference between Pst-treated and osmotically-stressed Pst-treated? I think these suggestions would strengthen the paper’s conclusions. However, if the authors have good reasons for not doing these, I think the paper is fit for publication without them.

Response: In the first version of the manuscript, we already conducted the statistical analysis in the way the reviewer is asking for. As example, the asterisk in Figure 1B meant that there is a significant difference between heat-stressed Pst-treated and Pst-treated, and between osmotically/heat-stressed Pst-treated and Pst-treated. The graphic also indicates that there is no difference between osmotically-stressed Pst-treated and Pst-treated.

Reviewer 2 Report

The manuscript presents the important findings in plant-pathogen interaction under multiple abiotic stresses. The experimental design is well designed, results and discussion are clear with scientific impact and the most important findings are well explained.

Study results confirmed the previous knowledge that combined abiotic and biotic stress cause the alteration in gene expression. After the application of heat, mannitol or their combination expression of pathogenesis-related proteins and TN-13 gene was inhibited which resulted in increase of disease susceptibility to Pseudomonas syringae. Similar was noticed for Botrytis cinerea disease resistance. In this part of research, mannitol-heat combination significantly reduced expression of several genes (PDF1.3, BOS1 and THI2.2). As the final output of study, authors revealed the alterations in expression of cell wall coding genes which are sensitive to abiotic-biotic stress interactions.

This study is worth publishing.

Author Response

Thank you very much.

Reviewer 3 Report

Authors investigated the effects of heat stress, osmotic stress and their combination on the susceptibility of plants to pathogen infection, and demonstrated that these stresses, especially combined stress, accelerated the effects of pathogens on plants. In addition, these enhanced susceptibility to pathogen by abiotic stresses might be due to the suppression of transcripts involved in defense and cell wall maintenance.

Unfortunately, the level of the research is below the standard of this journal. Additional experiments are required.

Basically, only pathogen response phenotype and gene expression will not provide any information. Yes, expression of pathogen response and cell wall maintenance genes were suppressed by stress combination, as well as each corresponding single stress. However, significance of these genes needs to be confirmed by the analyses of mutants. I know these transcripts tested in this study have already shown to be essential for the pathogen responses, but, roles of these genes under stress combination have not been investigated.

When we compared plants between Figure 1 and 2, effects of osmotic stress without pathogen infection seem to be different (plant seems to be much smaller in Fig 2 compared to Fig 1). Authors should explain why it happens. 

Author Response

Point 1: Basically, only pathogen response phenotype and gene expression will not provide any information. Yes, expression of pathogen response and cell wall maintenance genes were suppressed by stress combination, as well as each corresponding single stress. However, significance of these genes needs to be confirmed by the analyses of mutants. I know these transcripts tested in this study have already shown to be essential for the pathogen responses, but, roles of these genes under stress combination have not been investigated.

Response: In the revised manuscript we extended the discussion to include the point raised by the reviewer. We now discuss that future work should focus on the analysis of mutants and overexpressing lines from the selected genes, such as those analyzed in this work, to shed light on the role of these genes under abiotic stress and to further deepen our understanding on the nature of the interactions between combined abiotic and biotic stresses (Lines 352-355).

Point 2: When we compared plants between Figure 1 and 2, effects of osmotic stress without pathogen infection seem to be different (plant seems to be much smaller in Fig 2 compared to Fig 1). Authors should explain why it happens.

Response: The photographs show individual plants taken from a pool of plants used for the analysis. The experiments shown in Figure 1 and Figure 2 were conducted separately with different batches of plants and in three biological replicates. For each experiment all plants used were first grown together, then separated into 4 groups (C, O, H, O-H) and subjected to the corresponding treatments. Then each group was separated again into two groups: one of the groups was used for the biotic treatment, and the second group was used as mock control. Thus, we consider that the small difference in size observed in the individual plants shown in the photographs do not influence the outcome of the work as we compare osmotic-stressed samples with its corresponding osmotic-pathogen infected samples within each experiment. We now rephrase the “Growth Conditions and Abiotic Stress Treatments “ description of the Material and Methods sections to make this point clearer (Lines 361-376).

Round 2

Reviewer 3 Report

It is OK to consider mutant analyses as contents of the next paper. But, authors should note that it is not the best way to publish.